# Identification and Functional Analysis of *bZIP* Genes in Cotton Response to Drought Stress

**DOI:** 10.3390/ijms232314894

**Published:** 2022-11-28

**Authors:** Boyang Zhang, Cheng Feng, Lin Chen, Baoqi Li, Xianlong Zhang, Xiyan Yang

**Affiliations:** National Key Laboratory of Crop Genetic Improvement, Huazhong Agricultural University, Wuhan 430070, China

**Keywords:** bZIP, *Gossypium hirsutum*, DNA-binding site, dimerization properties, drought stress response, expression analysis, segmental duplication, tandem duplication

## Abstract

The basic leucine zipper (bZIP) transcription factors, which harbor a conserved bZIP domain composed of two regions, a DNA-binding basic region and a Leu Zipper region, operate as important switches of transcription networks in eukaryotes. However, this gene family has not been systematically characterized in cotton (*Gossypium hirsutum*). Here, we identified 197 bZIP family members in cotton. The chromosome distribution pattern indicates that the *GhbZIP* genes have undergone 53 genome-wide segmental and 7 tandem duplication events which contribute to the expansion of the cotton *bZIP* family. Phylogenetic analysis showed that cotton GhbZIP proteins cluster into 13 subfamilies, and homologous protein pairs showed similar characteristics. Inspection of the DNA-binding basic region and leucine repeat heptads within the bZIP domains indicated different DNA-binding site specificities as well as dimerization properties among different groups. Comprehensive expression analysis indicated the most highly and differentially expressed genes in root and leaf that might play significant roles in cotton response to drought stress. *GhABF3D* was identified as a highly and differentially expressed *bZIP* family gene in cotton leaf and root under drought stress treatments that likely controls drought stress responses in cotton. These data provide useful information for further functional analysis of the *GhbZIP* gene family and its potential application in crop improvement.

## 1. Introduction

Plants experience diversity in environmental factors during their lifecycles. When they suffer from abiotic stresses, they can evolve morphological, physiological, and biochemical adaptations, which are coordinated by the regulation of expression by some transcription factors (TFs) in order to optimize their growth and survival. TFs play important roles in alleviating plant damage under stresses by activating or repressing target genes [1]. The basic leucine zipper (bZIP) family is one of the largest groups of TFs in plants. It operates as important switches in the regulation of developmental and physiological processes as well as abiotic and biotic stress responses [2].

The family is defined by the bZIP domain, which comprises 60–80 amino acids and contains two functionally distinct regions: a DNA-binding basic region and a Leu Zipper region [3]. The basic region, with 16 amino acid residues, contains a nuclear localization signal and an N-x7-R/K motif that binds DNA. The leucine zipper region at the C-terminus contains a Leu or other hydrophobic amino acids (Ile, Val, Phe, or Met) every six amino residues, which enables DNA binding by associating with the acid domain to form homo- and/or heterodimerization [2,4,5,6,7,8]. bZIPs preferentially bind to DNA containing the ACGT core sequence, and their binding specificity is regulated by nearby nucleotides, mainly including an A-box (TACGTA), C-box (GACGTC), and G-box (CACGTG) [2,4,9,10].

The bZIP TF family is conserved across eukaryotes and has undergone intensive gene expansion in angiosperms [11]. As the sequencing of genomes for more species is completed, numerous plant bZIP proteins are being identified and characterized. The number of bZIP family member genes is known in several diploid angiosperms, such as 78 in Arabidopsis [4,10], 89 in rice [2], 92 in sorghum [5], 89 in barley [12], 125 in maize [6], and 131 in soybean [13]. Polyploids have undergone more duplication events, resulting in the larger number of bZIP members, for example, *bZIP* gene family members in rapeseed are 247 in total [14]. In angiosperms, bZIP TFs can be classified into 13 different subfamilies according to their DNA-binding specificity and domain similarity, namely, A, B, C, D, E, F, G, H, I, J, K, M, and S [2,10].

Considerable evidence shows that bZIP TFs take part in growth and developmental processes, as well as stress responses [10,15,16,17]. *AtbZIP14/FD* preferentially expressed in the shoot apex is required for *FLOWERING LOCUS T (FT)* to promote flowering [16]. The Arabidopsis transcription factor gene *AtbZIP53* regulates seed maturation by influencing heterodimerization and protein complex formation [18]. *AtbZIP1* is involved in sugar signaling, protein networking, and DNA binding [19]. *AtbZIP17* enhances salt tolerance when expressed by the stress-inducible promoter *RD29A* [20]. Overexpressing *OsbZIP23*, a central regulator of ABA signaling and biosynthesis, significantly improves drought and salt tolerance in rice [21]. In plants, group A bZIP TFs are mainly related to ABA and stress responses, among which the prominent subgroup of abscisic-acid-responsive-element-binding factors (ABFs) have been extensively investigated in many plant species, such as Arabidopsis, rice, and potato [22,23,24,25]. In Arabidopsis, more than half of group A bZIP TFs (AtbZIP39/ABI5, AtbZIP36/ABF2/AREB1, AtbZIP38/ABF4/AREB2, AtbZIP66/AREB3, AtbZIP40/GBF4, AtbZIP35/ABF1, and AtbZIP37/ABF3) have been well investigated, and most of those TFs acted as core elements in ABA and stress signals [10,23,26].

Even though many important functions have been recognized, and sequencing of the cotton genome has been completed [27] and improved [28], few detailed analyses have previously been reported for bZIP family proteins in cotton. Overexpression of *GhABF2*, which was induced by drought and ABA stress but repressed by high salinity, significantly improves drought and salt stress tolerance in Arabidopsis and cotton [29]. Overexpression of *GhABF2D* increases dehydration resilience and drought resilience through stomatal regulation [30].

In this study, 197 bZIP family genes were identified and characterized in upland cotton through sequence alignment on the basis of previous studies. The phylogenetic analysis, gene structure, conservative domain, and stress expression patterns were further studied. Moreover, the function of *GhbZIP177*, a differentially expressed bZIP gene, was characterized in the cotton response to drought stress. The GhbZIP protein may play a key role in regulating the abiotic stress response of cotton. Our research can provide useful clues for the subsequent analysis of the function and regulation mechanism of potentially important GhbZIP proteins.

## 2. Results

### 2.1. Identification of bZIP Transcription Factors in G. Hirsutum

In order to identify the members of *bZIP* transcription factors genes in cotton, we compared the cotton (*G. hirsutum*) protein sequences with the conservative domain hidden Markov model (HMM) of bZIP transcription factors downloaded from the Pfam website. After obtaining possible bZIP family members, we finally determined 197 cotton proteins as potential bZIPs through further SMART analysis (E value < 1.0), which were named as GhbZIP proteins. On the basis of the genomic position information obtained from the database, we showed that the *GhbZIP* genes were found across all 26 cotton chromosomes, ranging from 2 to 15 per chromosome (Figure 1A, Appendix A). The GhbZIP genes are unevenly distributed on each of the 26 cotton chromosomes, with certain chromosomes and chromosomal regions having a relatively high density. For instance, 15 GhbZIP genes are located on chromosomes A05, whereas there are only two presented on chromosome A04. We named these genes *GhbZIP1* to *GhbZIP197*, on the basis of their position on cotton chromosomes A1 to A12, D1 to D13, and from top to bottom (Figure 1B, Appendix A). Information on open reading frame (ORF) length, protein length, and chromosomal location of all 197 *GhbZIP* genes is listed in Appendix A.

To investigate the mechanisms underlying *GhbZIP* gene family evolution, both tandem and segmental duplication events were examined. It was found that the *GhbZIP* genes in upland cotton can produce a large number of family members by tandem duplication or fragment replication. Seven pairs of *GhbZIP* genes, with two on chromosome A05 and five from A09, A12, D05, D12, and D13 each were found to be arranged in tandem, with duplication genes adjacent to each other on the same chromosome (Figure 1B). Fifty-three pairs of *GhbZIP* genes were found to be located on the duplicated segmental regions of 14 cotton chromosomes, with all pairs of duplicated segments occurring in the same subgroups (Figure 1B). Surprisingly, eight pairs of *GhbZIP* genes are generated by duplication of chromosomal fragments from non-homologous subgenomes (A03 and D02) (Figure 1B). Overall, >60% of *bZIP* genes came from fragment duplication or gene duplication in upland cotton. These results suggest that both tandem duplication and segmental genome duplication events have contributed largely to the expansion of the bZIP transcription factor gene family in upland cotton.

We also analyzed *bZIP* family genes in other cotton species, i.e., *G. barbardense*, *G. arboreum*, and *G. raimondii* using the same procedure as for *G. hirsutism*. We identified 207, 107, and 112 *bZIP* family genes in *G. barbardense*, *G.arboreum*, and *G. raimondii*, respectively (Appendix A). Among the collinearity pairs of *bZIP* genes in the four cotton species, 82 and 91 pairs of *bZIP* genes displayed collinearity between *G. hirsutum* and *G. barbardense* with *G. raimondii*, respectively, among which 45 pairs of *bZIP* genes from *G. raimondii* formed a collinearity relationship with both G. *hirsutum* and *G. barbardense* (Figure 2A, Appendix A). Meanwhile, 103 and 102 pairs of *bZIP* genes showed collinearity between G. *hirsutum* and *G. barbardense* with *G. arboretum*, respectively, among which 54 *bZIP* genes from *G. arboretum* formed a collinearity relationship with both G. *hirsutum* and *G. barbardense* (Figure 2B, Appendix A).

### 2.2. Phylogenetic Analysis, Prediction of Conserved Motifs, and Display of Gene Structure

To evaluate the evolutionary relationships of 197 *GhbZIP* genes, we conducted a phylogenetic analysis on the basis of their full-length protein sequences. Applying the NJ method, we assigned the proteins to 13 main groups: A through K, M, and S (Figure 3), which is the same group number as those found in Arabidopsis [10]. Unlike the other groups in this phylogenetic tree, Subgroup B contained a single GhbZIP protein, and Subgroups H, K, J, and M contained a pair of homologous GhbZIP proteins (Figure 3).

The evolutionary relationships among these GhbZIP proteins were also determined by analyzing their conserved motifs. Protein structures of different group GhbZIP proteins were identified by MEME to research other possible conservative domains. These composition patterns tended to be consistent with the results from our phylogenetic tree, being nearly identical among proteins within the same group, but varying greatly between groups. This revealed that all of the potential bZIP transcription factors contain a leucine zipper region with heptad repeats and an invariant N-x7-R/K motif in the basic regions. Other than the conserved bZIP domains, fourteen other motifs were present only in certain groups, which might explain why functions for GhbZIP proteins tend to be specific to a particular group. These included Motifs 1 and 2, only present in Group A; Motif 3, in Group C; Motif 4, Group D; Motif 5, Group E; Motifs 6 and 7, Group F; Motifs 8 and 9, Groups G; Motif 10, mainly present in Group H; Motif 11, Group I; Motif 12, Group J; Motif 13, Group K; and Motif 14, Group S (Figure 4).

To further understand the structure of *GhbZIP* genes, we analyzed the intron phase in its conservative regions (Appendix A). We constructed the representative genes structures for the 13 groups. Fifty genes had no introns. It could be found that different bZIP genes in the same subgroup might also have different structures. Among them, the number of introns contained in Groups D and G was relatively large, up to 12. The number of introns contained in Subgroup S was small, and most of the bZIP genes in this group (45 genes, accounting for 93.75% in the group) had no introns. Moreover, genes in Groups A, C, E, and I mainly had 3–6 introns (Appendix A).

The intron phase of bZIP conservative domain region is shown in Figure 5A. On the basis of intron presence or absence, as well as position, 147 intron-containing *GhbZIP* genes were divided into four groups (Appendix A). Thirteen had no intron in the bZIP domain region (Figure 5B). Types 2 and 3 had one intron each, but the intron position was different, with one intron present in the hinge region (Figure 5C, Appendix A) or basic region (Figure 5D, Appendix A). Type 4 had two introns each in the hinge region and basic region. In each group, the *GhbZIP* genes were also divided into subgroups on the basis of the intron position and splicing phase (P0, P1, or P2). Compared to the splicing phase in rice [30], the splicing phases in cotton were more complex.

### 2.3. DNA-Binding Site Specificity Prediciton of GhbZIP Transcription Factors

bZIP transcription factors are considered to bind DNA sequence elements by means of the basic region of 16 amino acid residues with an invariant N-x7-R/K motif in their bZIP domain. They preferentially bind to DNA sequences with an ACGT core, which include three groups, the A-box (TACGTA), C-box (GACGTC), and G-box (CACGTG) [9].

To predict the DNA-binding site specificity of the GhbZIP proteins, 197 bZIP proteins can be combined into 10 groups according to the basic and hinge amino acid characteristics (Table 1, Appendix A). Putative DNA-binding sites were identified in eight groups (except E and J groups), with six (B-D, F, G/M/S, H) having explicit G-box or C-box binding sites, suggesting that the G-box or C-box sites are the main DNA-binding sites for bZIP members in upland cotton. Group A (covering 41 members) displays an ABRE binding site, a critical binding site responding to ABA signals. Group I (including 25 members) has DNA sequences such as CCA/TGG repeats other than the ACGT core. The results show that the variable binding sites or binding affinity for different bZIP members is determined by the certain key amino acid residues in the conservative domain of bZIP proteins. Hence, it is crucial to predict and verify the DNA-binding ability of *GhbZIP* genes.

### 2.4. Dimerization Properties Prediction of bZIP Transcription Factors

The leucine zipper region of the bZIP transcription factor is composed of heptad repeats, whose lengths were variable from two to nine in a previous study. In this study, we also found similar results in GhbZIP proteins (Appendix A). As in rice, the positions of the amino acids are named as *g*, *a*, *b*, *c*, *d*, *e*, and *f* in that order in the heptad [2]. The stability and specificity of the Leu zipper dimerization are determined by the amino acids near it’s a, *d*, *e,* and *g* positions [2]. Thus, a detailed analysis was conducted to characterize the amino acids present at the *a*, *d*, *e*, and *g* positions.

About 25% of amino acids present at the *a* position are Asn (Figure 6A), which is almost equal to the frequency observed in rice (23%) and Arabidopsis (22%) [2,8]. The *a* position Asns are considered to form homodimers by stable N–N interactions at the *a* ↔ *a*′ pairs. For each heptad, the highest Asn frequency at the *a* position was located in the second heptad followed by the fifth heptad, accounting for 58.9 and 57.4 %, respectively (Figure 6B), as observed earlier for AtbZIP proteins [8]. This indicates that a larger number of GhbZIPs may prefer to homodimerize. Other hydrophobic amino acids (I, V, L, and M) were also found in the *a* position of some heptads in some GhbZIPs. A small number (16%) of charged amino acids (K, R, D, and E) present at the *a* position were observed, which might contribute to hetero-dimer formation. Not surprisingly, the frequency of Leu in the *d* position in GhbZIPs was found to be 59% (Figure 6A), which was slightly higher than that in AtbZIPs (56%); however, the abundance of other aliphatic amino acids (15%) was less than that in AtbZIPs (19%), which indicated higher dimer stability of longer zippers in cotton.

The *e* and *g* positions typically contain charged residues (acidic amino acids E and D, and basic amino acids R and K), which provide attractive or repulsive pairing in each heptad [31]. The results showed that GhbZIPs had a low frequency of charged amino acids at the e position (28%), while a comparable frequency in *g* positions (54%), compared to AtbZIPs (41% in *e*, 53% in *g*) (Figure 6A).

As a result, in each heptad of GhbZIP Leu zippers, the frequency of attractive or repulsive *g* ↔ *e*′ pairs was calculated for four types: repulsive basic pairs (such as K ↔ K, R ↔ K, and R ↔ R), repulsive acidic pairs (such as E ↔ E, E ↔ D, and E ↔ Q), attractive basic–acidic pairs (R ↔ E and K ↔ E), and attractive acidic–basic pairs (E ↔ R, E ↔ K, D ↔ R, and D ↔ K). The histogram of their frequency is presented in Figure 6C. It was observed that the highest frequency of interactive *g* ↔ *e*′ pairs was 42.64%, which was present in the first heptads and decreased sharply in the next heptads. It should be noted that multiple repulsive *g* ↔ *e*′ pairs were absent in GhbZIP proteins (Appendix A). When the *g* ↔ *e*′ pairs are repulsive pairs, they favor the formation of heterodimerization. When the *g* ↔ *e*′ pairs are attractive pairs, self-complementary salt bridges form to block the formation of homo-dimers or hetero-dimers. GhbZIP proteins are expected to display complex and varied dimerization patterns, with the potential to form homodimerization and/or heterodimerization patterns.

### 2.5. Expression Analysis of GhbZIP Family Members in Cotton under Drought Treatments

The available transcriptome data sets have made it possible to comprehensively analyze *bZIP* genes expressed during development processes in cotton. We collected the transcriptome data sets from different tissues/organs, including the root, stem, leaf, petal, anther, stigma, ovule, seed, and fiber (Appendix A), and identified 185 (93.9%) *GhbZIP* genes expressed (FPKM ≥ 1) at least in one tissue, with 76 (38.6%) genes expressed in all tissues (FPKM ≥ 1), and with 21 and 23 *GhbZIP* genes showing their highest expression levels (FPKM ≥ 20) in root and leaf, respectively (Figure 7A,B). A total of 17 *GhbZIP* genes were highly expressed in root and 12 in leaf (Figure 7C,D).

In order to analyze the physiological functions of GhbZIP genes involved in drought response in cotton, we conducted drought treatments on the seedling stages of drought-sensitive variety ZY7 and drought-tolerant variety ZY168 and then analyzed their expression in roots and leaves. The results show that the expression levels of all *GhbZIP* genes changed during drought stress. Most of the *GhbZIP* genes displayed different expression patterns in drought-sensitive ZY7 and drought-tolerant ZY168 in roots, while more *GhbZIPs* genes displayed similar expression patterns in leaves in both genotypes (Appendix A). Among them, 38 *GhbZIP* genes, with 17 and 21 members from Group A and S (representing 46.9%), were differentially expressed either in root or leaves, either in drought-sensitive ZY7 and drought-tolerant ZY168 under drought stress (Figure 7F). Among the 17 Group A members, three homologues of Arabidopsis *FD*, namely, *GhbZIP16*, *GhbZIP122*, and *GhbZIP130*, showed decreased expression either in ZY7 or ZY168, while another homologue *GhbZIP173* was upregulated in ZY7 and downregulated in ZY168. *GhbZIP20,* one homologue of Arabidopsis *ABI5,* was downregulated in ZY7 both in roots and leaves. However, it showed an opposite expression pattern in roots and leaves in ZY168. Two *ABF2* homologous gene pairs, *GhbZIP36* and *GhbZIP134*, both showed upregulation under drought stress in both genotypes. However, they showed different upregulatory patterns in roots and leaves. Another gene pair, *GhbZIP79* and *GhbZIP177*, showed homology with *AtABF3.* They displayed similar regulatory patterns in ZY7 (both upregulated in leaves and roots). However, *GhbZIP177* was also upregulated in roots and leaves in ZY168, while the expression of *GhbZIP79* increased in leaves and decreased in roots. Seven *TGA* homologous Group D members, namely, *GhbZIP64*, *GhbZIP107*, *GhbZIP129*, *GhbZIP143*, *GhbZIP147*, *GhbZIP162*, and *GhbZIP189*, displayed different expression patterns in drought-sensitive ZY7 and drought-tolerant ZY168, as well as in roots and leaves (Figure 7F).

### 2.6. GhbZIP177 Conferred Drought Resistance in Cotton

From the above analysis, *GhbZIP177* (Ghir_D12G002440, *GhABF3D*) was found to be preferentially expressed both in root and leaf (Figure 7D,E), and it was differentially expressed during drought treatments in cotton (Figure 7F). The full-length CDS of *GhbZIP177* is 1281 nucleotides and encodes a protein of 426 amino acids (Appendix A). Phylogenetic analysis showed that GhbZIP177 is closely related to Arabidopsis AtbZIP37/AtABF3, a Group A bZIP protein (Figure 3). Subcellular localization in *N. benthamiana* leaf epidermis indicated that GhbZIP177 is localized to the nucleus (Figure 8A).

In order to study the function of *GhbZIP177* in cotton under drought conditions, several overexpression and RNA interference (RNAi) transgenic cotton lines were developed. Two RNAi lines (ARi9 and Ari11) and two overexpression lines (AOE6 and AOE9) were selected for further study. A null line (negative plants segregated from AOE6) was used as the control. The expression levels of *GhbZIP177* in these transgenic lines were verified by RT-PCR and qRT-PCR (Figure 8B,C). The expression levels of *GhbZIP177* were reduced to 16.0% and 17.8% of the control in ARi9 and ARi11, respectively, and increased up to 6.1- and 14.3-fold compared to control in overexpressing lines AOE6 and AOE9, respectively (Figure 8C).

To investigate drought stress tolerance of these transgenic cotton lines, trefoil stage seedings were exposed to drought stress by withholding water for 14 days. The results show that AOE6 and AOE9 plants exhibited enhanced drought tolerance compared to null, while ARi9 and ARi11 wilted prematurely after 10 days withdrawal of water (Figure 8D). The plants were re-watered after 14 days withdrawal of water. Moreover, AOE6 and AOE9 were restored after 4 and 9 days of re-watering (Figure 8D), and almost all the RNAi plants died, while >85% of the overexpressing plants survived, and about 60% null plants survived after 9 days of re-watering (Figure 8E). We further analyzed the water loss in null and transgenic plants during dehydration by comparing the water content of excised leaves. More water loss was evident in ARi9 and ARi11 than in null plants, and plants overexpressing *GhbZIP177* showed decreased water loss of excised leaves (Figure 8F). These results suggest that *GhbZIP177* positively regulates drought stress tolerance in cotton.

## 3. Discussion

Transcription factors can specifically bind to cis-acting elements to precisely regulate gene expression. The basic leucine zipper (bZIP) is one of the largest groups of TFs in plants, containing two functionally distinct regions: a bind DNA-binding region and a Leu Zipper region. Recent research has confirmed that the bZIP family can play important roles in regulating growth and development as well as in responses to biotic and abiotic stress [32,33,34].

Recently, with the successful sequencing of various plant genomes, systematical analyzing of *bZIP* family genes are down in eukaryote plants, such as *Arabidopsis thaliana* (*n* = 78) [11], rice (*n* = 89) [2], maize (*n* = 125) [35], poplar [36], and soybean (*n* = 131) [37]. However, there are few systematic studies on the cotton bZIP gene family besides the work done by Wang et al. [38]. In their study, they identified 207 GHbZIP proteins on the basis of HMM profile using a reference genome (ZJU) [39]. Similar to their results, the cotton bZIP family could be divided into 13 subfamilies as in Arabidopsis [11]. However, in the current study, we used the reference genome published by our laboratory [28]. After the blast search and domain search, we also manually checked each protein to determine the final number of GhbZIPs. Moreover, we also analyzed the duplication events between intra-subgenomic, inter-subgenomic, and inter-genomic levels. Upland cotton is allotetraploid, and its formation has undergone a complex process of hybridization, polyploidy, and domestication. It was formed by hybridization and genome doubling of diploid cotton from Africa and diploid cotton from America [30]. The results show that >60% bZIP genes came from fragment duplication or gene duplication in upland cotton, which is consistent with the proportion of fragment duplication events or gene duplication events in soybean and rice, but less than in poplar [36].

The *bZIP* gene family was highly conserved, with all members containing motif 1, with additional conservative motifs that were also clustered according to the classification of subgroups. The analysis of gene structures also showed a similar phenomenon, whereby *bZIP* genes of the same subgroup had similar gene structures. These results suggest that *bZIP* of the same subgroup might have similar functions, while *bZIP* genes among subgroups might have different functions, and might play specific roles. As in previous studies [2,10], most genes in Subgroup S were intronless. Intronless genes are a characteristic feature of prokaryotes. Introns are beneficial to the evolution of species, which increase the length of genes, improve the frequency of intergenomic recombination, and play multiple regulatory roles, while intronless genes have no advantages for species evolution and recombination [40]. Studies have shown that genes with fast response under stresses are more likely the genes without introns or genes with few introns [41]. In accordance with this, we found that more genes from Group A (3–6 introns) and S (intronless) showed differential expression under drought stress (Figure 7). These might suggest that these those *bZIP* genes might play more important roles during drought stress in cotton.

The DNA binding specificity of bZIP proteins is determined by specific amino acid sequences of the most conservative basic region and hinge region in the domain, which can directly interact with DNA cis-acting elements [42]. bZIP transcription factors have a binding reference for ACGT core sequences, such as A-box, C-box, and G-box in plants [4,10]. Our results found that most bZIP members contain G-box or C-box binding sites, which is consistent with the results for other plants [2,10]. bZIP proteins could form homo- and/or heterodimers. The amino acid residues at *a*, *d*, *e*, and *g* positions may play an important role in the oligomerization regulation of leucine zipper region, as well as the specificity and stability of dimers [6]. In this study, we analyzed the amino acid residues at *a*, *d*, *e*, and *g* positions and the electrification of *g* ↔ *e*′ pairs, and the differences and similarities between bZIP dimerization in upland cotton and other species were discussed, indicating the uniqueness and complexity of bZIP gene dimerization in upland cotton.

In cotton, Group A bZIP TFs consist of 55 members, representing the second biggest group. The prominent subgroup of ABFs containing ABF1-4 has been found to act as important patterners of ABA and stress signals in plants [22,23,24,25]. In particular, *ABF3* is induced by ABA and osmotic stresses such as water deficient and high salinity [43,44,45,46]. Arabidopsis *AtABF3* functions redundantly with *AtABF2* and *AtABF4* as master regulators of ABA signaling in response to drought and osmotic stresses, although their functions do not completely overlap [44]. Liang et al. [29] reported a Group A gene, *GhABF2*, which was induced by drought and ABA stress but repressed by high salinity. They found that GhABF2 overexpression cottons exhibited increased fiber yields under drought and saline wetland. In the current study, the function of *GhbZIP177* (*GhABF3*) in cotton responsive to drought stress was investigated. Overexpression of *GhbZIP177* could improve the survival rate while decreasing the expression of *GhbZIP177,* reducing the survival rate after drought treatment and re-watering. However, the mechanism needs further study.

## 4. Materials and Method

### 4.1. Identification of bZIP Family Genes in Gossypium spp

To identify possible bZIP transcription factor genes in four cotton species (*G. hirsutum*, *G. arboreum*, *G. barbadense*, and *G. raimondii*), we download all the protein sequences from CottonFGD (https://cottonfgd.net/about/download.html accessed on 21 November 2021) [27,28,47]. Meanwhile, we downloaded all 78 AtbZIPs family protein sequences from the TAIR website (https://www.arabidopsis.org/ accessed on 21 November 2021) according to known research reports. A BLAST-P search (E value, 1e-5) of all the annotated cotton proteins was carried out using the HMM profile of the AtbZIPs domain as a query. Then, we used SMART (http://smart.embl-heidelberg.de/ accessed on 21 November 2021) [48] to determine whether the potential bZIP genes obtained in the previous step contained bZIP domains.

### 4.2. Chromosomal Locations and Gene Duplication Analysis

In order to analyze the duplication and tandem duplication events of the *bZIP* genes in *G. hirsutum*, we downloaded the cotton genome annotation file and bZIP amino acid sequence of bZIP from CottonFGD. The duplication and tandem repeat files of the bZIP genes were obtained by MCScanx software [49], and finally the results were visualized by the Gene Location Visualize from the GTF/GFF program in TBtools [50].

### 4.3. Phylogeny Analysis bZIP

Multiple sequence alignments of the protein sequences were carried out using ClustalX (ver.1.83) with default settings. The complete amino acid sequences of the predicted bZIP proteins were used for phylogenetic analysis. Phylogenetic trees were constructed using MEGAX software by the Neighbor-Joining method [51]. The evolutionary distances were computed using the Poisson correction method, and the nodes of the trees were evaluated by boot-strap analysis with 1000 replicates. Finally, the iTOL website (https://itol.embl.de/ accessed on 2 December 2021) was used to refine the phylogenetic tree [52].

### 4.4. Sequence Analysis

The identification of additional conserved domains of 197 bZIP proteins was conducted by MEME software (https://meme-suite.org/meme/doc/meme.html accessed on 2 December 2021), version 5.4.1, with all parameters set to default. According to the specificity of their DNA-binding sites, these motifs are significant when they are shared by most of the GhbZIP proteins in the same group. To obtain the intron/exon structure of the bZIP genes, we downloaded the genome annotation file and used the Gene Structure View in TBtools for visualization and then merged the *bZIP* genes with a similar gene structure in each subgroup together.

### 4.5. Expression Profiles

The transcriptome data sets corresponding to expression abundances of TM-1 (the allotetraploid cotton *G. hirsutum L*. acc. Texas Marker-1) in different tissues and stresses from NCBI were used in the analysis of the expression profiles of GhbZIP genes (https://www.ncbi.nlm.nih.gov/sra/?term=PRJNA248163 accessed on 25 March 2021) as well as CottonFGD. The gene expression patterns were shown by pheatmap with the expression values normalized by ‘scale row’. The stress expression profile data were obtained from the data of two tissues (leaf and root) at seedling stages before and after drought treatment of the drought-sensitive material ZY7 and the drought-tolerant material ZY168 identified in our previous study [53].

### 4.6. Functional Analysis of GhbZIP177 in Cotton Response to Drought Stress

The coding sequence and RNAi fragments of *GhbZIP177* (Ghir_D12G002440) were cloned and inserted into the vectors pK2GW7 and pHellsgate4, respectively, using Gateway cloning technology in order to generate the overexpression vector *GhbZIP177*-pK2GW7 and RNAi vector *GhbZIP177*-pHellsgate4. As previously mentioned, transgenic plants are produced through Agrobacterium-mediated transformation (54). Using GhUBQ7 as an internal reference gene, the expression of *GhbZIP177* in transgenic plants was verified by RT-PCR and qRT-PCR methods in cotton leaves.

For subcellular localization, full-length *GhbZIP177* was cloned into the vector pGWB741 with GFP fused to the C-terminus. The *35S_pro_:GhbZIP177-GFP* construct was transiently expressed in tobacco epidermal cells following *Agrobacterium*-mediated transfection. The pGWB741 empty vector harboring *35S_pro_:GFP* was used as the control. GFP fluorescence was detected under a confocal microscope (Olympus FV1200) at two days following transfection.

T_3_ generations of RNAi, overexpression, and null lines were grown in controlled environment rooms at 25 °C with a 16 h light/8 h dark photo-period. Three-leaf stage plants were used to perform drought treatment with natural drought for 14 days. Photos were taken after natural drought for 10 days. The water loss rate and survival rate were calculated as described previously [54]. Three biological replicates were performed in each assay.

## Figures and Tables

**Figure 1 ijms-23-14894-f001:**
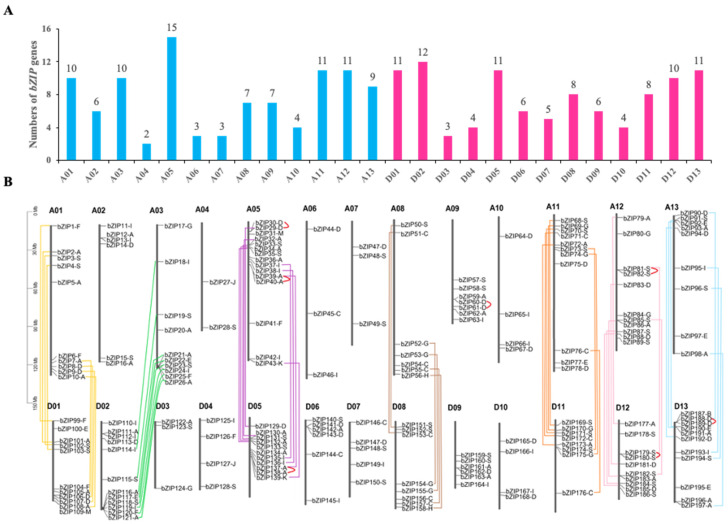
Distribution of 197 *GhbZIP* genes on cotton chromosomes. (**A**) Distribution of *GhbZIP* genes on different cotton chromosomes. (**B**) Duplication events of *GhbZIPs* on cotton chromosomes. *GhbZIP* genes are numbered 1 to 197. The duplication events of cotton *GhbZIP* genes are presented in color, of which 7 tandem duplications are expressed in red lines, and the replication of segments on different chromosomes are expressed in other colors.

**Figure 2 ijms-23-14894-f002:**
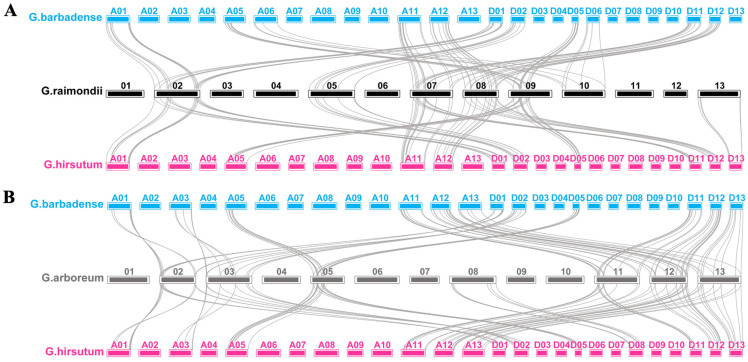
Collinearity analysis of *bZIP* genes between diploid and tetraploid cotton. (**A**) Collinearity analysis between *G. hirsutum*, *G. barbardense*, and *G. raimondii*. (**B**) Collinearity analysis between *G. hirsutum*, *G. barbardense*, and *G. arboretum*. Blue, magenta, black, and gray boxes represent chromosomes in *G. barbardense*, *G. hirsutum*, *G. raimondii*, and *G. arboretum*, respectively.

**Figure 3 ijms-23-14894-f003:**
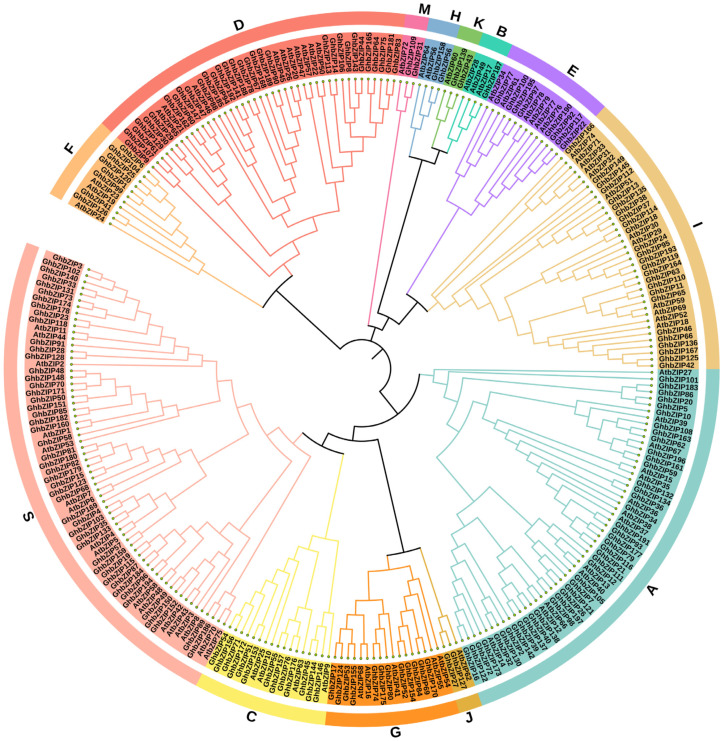
Phylogenetic analysis and subgroup classification of GhbZIP proteins. The phylogenetic tree of bZIP proteins was performed on the basis of similarities in the conserved bZIP domains and the homologues proteins in Arabidopsis. The 197 bZIP proteins (bZIP1 to bZIP197) were classified into 13 groups (A through K, M, and S), following the generic naming according to their position on chromosomes.

**Figure 4 ijms-23-14894-f004:**
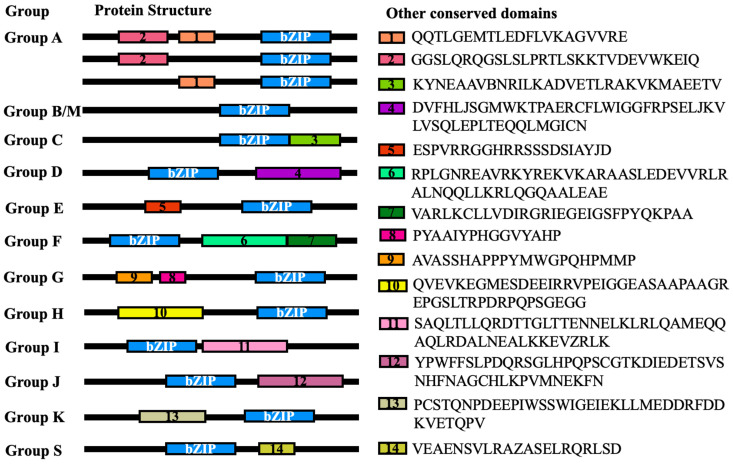
Conserved bZIP domains and additional conserved motifs present in GhbZIP proteins in different groups. The bZIP domains are shown in blue. Other conserved motifs outside the bZIP domain are highlighted in different color boxes with numbers 1 to 14, with details of conserved motifs on the right side.

**Figure 5 ijms-23-14894-f005:**
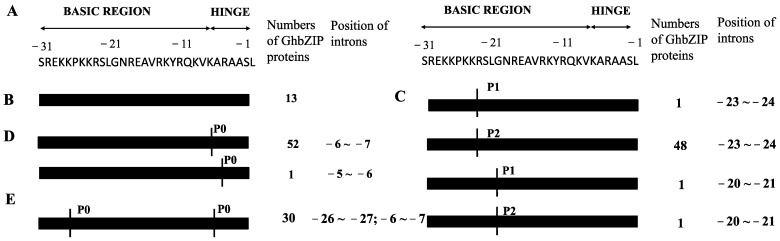
Intron patterns within the basic and hinge regions of GhbZIP proteins. P0, P1, and P2 stand for the intron splicing phases. P0, P1, and P2 represent the splicing occurring after the third, second, and first nucleotides of the codon, respectively. The black bars represent the sequence of basic and hinge regions. The vertical lines denote the positions of intron splicing phases. (**A**) The representative amino acid of the conserved bZIP domain. (**B**) Thirteen GhbZIP proteins had no intron in the bZIP domain. (**C**) GhbZIP proteins had one intron present in the hinge region. (**D**) GhbZIP proteins had one intron present in the basic region. (**E**) GhbZIP proteins had two introns each in the hinge region and basic region.

**Figure 6 ijms-23-14894-f006:**
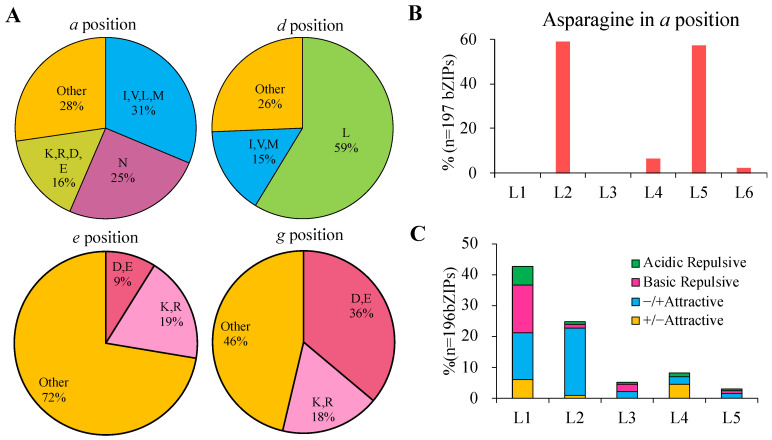
Prediction of the dimerization properties of 197 GhbZIP proteins. (**A**) Frequency diagram of different amino acids at *a*, *d*, *e,* and *g* positions in heptads of GhbZIP protein. (**B**) The frequency of Asn residues present in GhbZIP proteins at the *a* position in each heptad. (**C**) The frequency of attractive or repulsive *g* ↔ *e*′ pairs in each heptad in GhbZIP proteins.

**Figure 7 ijms-23-14894-f007:**
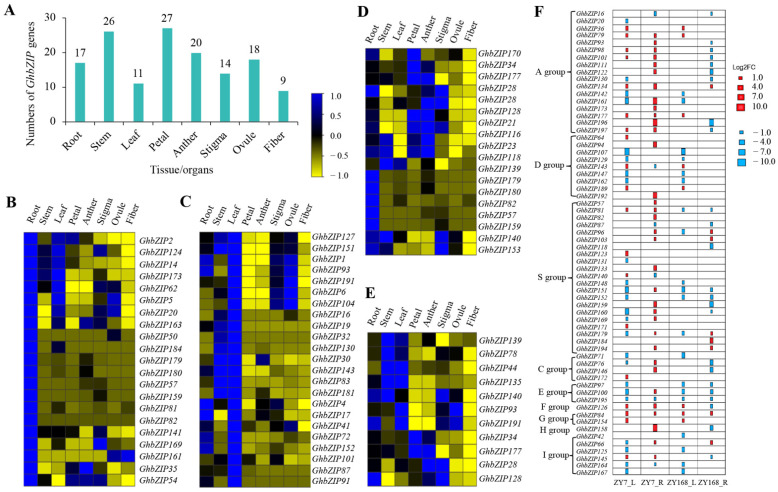
Expression analysis of *GhbZIP* genes. (**A**) The number of highly expressed *GhbZIP* genes (FPKM ≥ 20) in different tissues/organs. The expression of *GhbZIP* family members in different tissues/organs were investigated using transcriptome datasets of *G. hirsutum*. (**B**) Heat-map of 21 preferentially expressed *GhbZIP* genes in root. (**C**) Heat-map of 23 preferentially expressed *GhbZIP* genes in leaf. (**D**) Heat-map of 17 highly expressed *GhbZIP* genes in root. (**E**) Heat-map of 12 highly expressed *GhbZIP* genes in root. (**F**) Heat-map of 64 differentially expressed *GhbZIP* genes under drought stress in leaf and root of seedling stage in drought-sensitive variety ZY7 and drought-tolerant variety ZY168.

**Figure 8 ijms-23-14894-f008:**
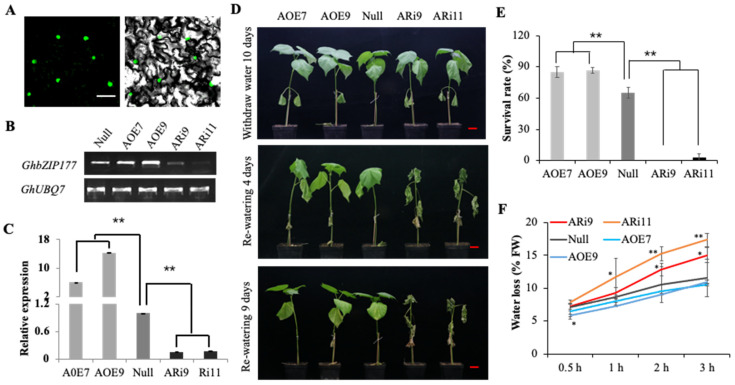
Functional analysis of *GhbZIP177* under drought stress treatments. (**A**) Subcellular localization of GhbZIP177 in *N. benthamiana* leaf epidermis. Bars = 50 μm. (**B**,**C**) Expression analysis of *GhbZIP177* in transgenic lines in cotton. The values represent means ± SE (*n* = 3). Significant difference analysis was calculated by LSD tests (** *p*-value < 0.01). (**D**) Phenotypes of transgenic and WT cotton plants under drought stress treatments. Plants at trefoil stage in soil were exposed to drought stress for 14 days and then re-watering. The photos were taken at 10 days withdrawing water, and 4 and 9 days after re-watering. Bars = 3 cm. (**E**) Survival rate of transgenic and WT plants after 7 days after re-watering. The values represent means ± SE (*n* = 3). Significant difference analysis was conducted by LSD tests (** *p*-value < 0.01). (**F**) Water loss of excised leaves from transgenic and WT plants at room temperature after 0.5, 1, 2, and 3 h. The values represent means ± SE (*n* = 5). Significant difference analysis was conducted by LSD tests (* *p*-value < 0.05, ** *p*-value < 0.01).

**Table 1 ijms-23-14894-t001:** Predicted DNA binding site of GhbZIP transcription factors.

Group	No. of Members	Characteristic Features of bZIP Domain	Putative Binding Site
A	41	*********MIK************QAY***	ABREs with the core ACGT or others containing GCGT/AAGT
B	1	**********RNRESAQLSR**********	G- and C-boxes with equal affinity
C	13	***********************QAHLEE*	Hybrid ACGT elements such as G/C, G/A, and C/G boxes
D	32	********LAQN**AA*KSR***KAYVQQ*	C-box sequence
E	8	*********A**************QYIS(/A)E*	Unknown
F/K	10	**************A****************	C-box elements
G/M/S	62	***********N**SA**SR**********	G-box and/or G-box-like sequences
H	2	***********NRVSAQQAR**********	G-box-like sequences
I	25	*******************K**********	Sequences other than those containing a palindromic ACGT core, such as CCA/TGG repeats
J	2	*******************I**********	Unknown

* represents un-conserved amino acids in the bZIP domain.

## Data Availability

Transcriptomic data were downloaded from the National Center for Biotechnology Information Sequence Read Archive (NCBI SRA: https://www.ncbi.nlm.nih.gov/sra accessed on 20 October 2022) under the bioproject PRJNA890393.

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
