# Peer review of "Identification and Functional Analysis of bZIP Genes in Cotton Response to Drought Stress"

_ijms, 2022, doi:10.3390/ijms232314894_

Round 1

Reviewer 1 Report

The research paper titled, “Identification and functional analysis of bZIP genes in cotton 2 response to drought stress”, reports of an important Transcription factor bZIP which is being intensively studied for their function in different species. In this paper, authors performed a genome-wide bZIP TF investigation on cotton (Gossipium hirsutum) and elucidated their function in abiotic stress.

However, the paper needs serious review and answer following comments.

Line 52-53 the sentence is not clear

Line 58. “Stress responses” might be unclear to some, using different word such as “stresses’ responses” is recommended.

Line 59. The full form of the abbreviation “FT” should be stated.

Line 60-63. Style used in naming genes should be consistent all over the manuscript, use Italic form for all genes.

Line 68-69. “Water stress” can indicate both drought and waterlogging. A clear phrase should be used.

Line 92. A published paper with same gene family (Wang et al., 2020) reported 207 GhbZIP genes, please state the difference comparing your findings  to theirs.

Line 104-108. The legend of figure 1 is not fully explained. Legend should describe “A” and “B” separately. Authors indicated “number in parentheses” but no parentheses are shown in the figure. In Figure 1 (B), gene names are not consistent with the names stated in the manuscript. Authors should also explain the meaning of alphabets given after the gene names. The large number of genes and lines are intermingled with each other, I advise to supplement the figure with table listing duplicated gene pairs and duplication type. A blue line shown besides bZIP124-G can be erased or linked with respective gene.

 Line 37. Correct the order of colors. G. hirsutum is given in magenta not blue.

Line 142. The sentence “A through K, L and S ….” Is not clear. There is no “L” subgroup in figure 3.

Line 144. Please clarify bZIP as either AtbZIP or GhbZIP proteins.

Line 167. Please correct the grammar error “…of different group GhbZIP proteins” to for example, “… present in GhbZIP proteins on different group”.

Line 173. Correct the grammar error “…have may also have…”.

Line 174-176. Authors should use consistence verb tense. Inconsistence use of ‘have’ and ‘had’ is confusing to reader.

Line 184. Please cite the reference stated regarding rice.

Figure 5. (A) Correct the spelling error “Basic refion”. Some letters are highlighted, please explain their meaning. (B) Authors mentioned 19 proteins in legend, but 15 is stated in figure.

 Line 198. Please correct the spelling error “Thiey”.

Line 258. An extra g is present and removed if made by error.

Figure 7. (A) The axis title “numbers” is not clear, please change to better title. I recommend writing the number of genes on top of each bar. (B, C, D, E) figures have no legend.

Line 274. Please rewrite the caption and state the meaning of abbreviations.

Figure 8. “,,in cotton” should be replaced with “in N. benthamiana/ tobacco”. (D) no bar is shown.

Line 326. The figure cited is referring to Figure 8C not 8D, please correct. 

 Line 328-331. The paragraph is not clear and leads to confusion, please rewrite according to supplied figure 8D.

Line 380. Please cite the references of other plants.

Discussion.

Line 365-374. Authors must compare their findings with other literature. The authors have repeated the writings as given in results.

The authors have not discussed their findings in GhbZIP177 over expressed transgenic plants.

Material and method.

Authors should cite literature of all software and webtools used in the MS (such as; genome of cotton, TAIR, BLAST, SMART,  MCScanx, TBtools, ClustalX, MEGAX, iTOL, MEME tool, etc.)

Wang, X., Lu, X., Malik, W. A., Chen, X., Wang, J., Wang, D., . . . Ye, W. (2020). Differentially expressed bZIP transcription factors confer multi-tolerances in Gossypium hirsutum L. International Journal of Biological Macromolecules, 146, 569-578. doi:https://doi.org/10.1016/j.ijbiomac.2020.01.013

Reviewer 2 Report

This manuscript has characterized cotton bZIP transcription factors at chromosome level, checked their expression pattern in each tissue and responsiveness to drought stress. This is a well-organized manuscript with detailed analysis and appropriate description. Here, I raised some minor points to be addressed or corrected before publication as below.

Lines 186-187. Please cite a reference showing the splicing phase in rice.

Please add a color key to Figure 7BCDE.

Please add statistics to Figure 8CF.

I think you have misspelled the gene name for GhbZIP177 in lines 312 and 323, where you wrote GhABF1. I think both names should be unified to GhbZIP177.

In “3. Discussion” section, I strongly recommend to add a novel discussion about responsiveness of GhbZIP177 to drought stress.  

Round 2

Reviewer 1 Report

Thank you for your corrections. It can be published as such.